# Community health knowledge and access to care in post-conflict Northern Uganda: Perspectives of community health workers in Pader District

Brett R. Albee[1]‡, Patrick Kasagara Atiya[2]‡, Otema Denish[2], Olanya Denish[2], Isaac V. Faustino[3], Dhatri Abeyaratne[4], Shayna D. Cunningham[5], Rogie Royce Carandang[6], Felix Bongomin[7]‡, Daniel S. Ebbs[8]*

1 School of Public Health, Yale University, New Haven, Connecticut, United States of America, 2 Northern Uganda Medical Mission, Pader, Uganda, 3 Department of Emergency Medicine, Yale School of Medicine, New Haven, Connecticut, United States of America, 4 School of Medicine, Yale University, New Haven, Connecticut, United States of America, 5 Department of Public Health Sciences, University of Connecticut School of Medicine, Farmington, Connecticut, United States of America, 6 Department of Health, Behavior, and Society, The University of Texas at San Antonio School of Public Health, San Antonio, Texas, United States of America, 7 Department of Medical Microbiology and Immunology, Faculty of Medicine, Gulu University, Gulu, Uganda, 8 Department of Pediatrics, Yale University School of Medicine, New Haven, Connecticut, United States of America

‡ This author are co-first author on this work. FB are co senior author on this work.
* Daniel.ebbs@yale.edu

## Abstract

Northern Uganda continues to experience high disease burdens and poor health outcomes shaped by poverty, geographic isolation, and long-standing health system constraints. Community health workers (CHWs) play an important role in rural districts by linking households to formal health services. In this study, CHWs refer to Village Health Team (VHT) members trained through the Laro Kwo Project in Pader District. However, limited research has examined how CHWs perceive community health priorities, barriers to care, and gaps between health knowledge and access. Understanding these perspectives is essential for designing responsive and sustainable community health programs. A qualitative descriptive study guided by interpretive principles was conducted across Pader District, Northern Uganda. Six focus group discussions were held between July 15 and 22, 2024 with 46 CHWs from six sub-counties, using a semi-structured guide covering eight domains related to community health, prevention practices, and program improvement. The guide was adapted from prior CHW-focused work and refined collaboratively with local partners. Data were documented through detailed field notes, translated, and analyzed inductively through iterative coding and thematic synthesis. Three overarching themes emerged: (1) experiencing health burden and community need, (2) barriers to access and systemic constraints, and (3) bridging health knowledge and everyday practice. CHWs identified malaria, maternal and child health complications, and a growing burden of non-communicable diseases as major community concerns. Persistent

**Data availability statement:** Data and manuscript draft are available through open science framework. Ebbs, D. (2025, September 6). Exploring Local Health Knowledge and Access: Focus Group Findings from Community Health Workers in Pader District, Uganda. Retrieved from osf.io/d2jgr.

**Funding:** This study was partially funded by the Yale School of Public Health Summer Internship funding award 2024 (BA) and funded by the Yale University Pediatric Scholars Program (DE) and Laerdal Foundation (DE). The funders had no role in study design, data collection and analysis, decision to publish, or preparation of the manuscript.

**Competing interests:** The authors have declared that no competing interests exist.

barriers to care included long distances to health facilities, medicine stockouts, and limited transportation. While communities demonstrated substantial knowledge of disease symptoms and prevention strategies, financial hardship and service limitations often prevented timely care-seeking and preventive action. CHWs' perspectives highlight a persistent gap between local health knowledge and access to care. Their recommendations emphasized priority health concerns and underscored the need for interventions that better align community knowledge with reliable and accessible services.

## Introduction

Rural and post-conflict regions across sub-Saharan Africa continue to face entrenched health disparities driven by poverty, geographic isolation, and the long-term impacts of political instability. Despite national and global efforts to expand access to healthcare and disease prevention, these gains remain unevenly distributed. In many resource-constrained settings, inequities are compounded by weak infrastructure, limited financing, and chronic shortages of trained health personnel, leaving many communities dependent on under-resourced local providers for essential services [1,2].

Northern Uganda exemplifies these challenges, particularly in districts like Pader, which report some of the country's poorest health outcomes despite national progress [3]. Decades of armed conflict, most notably the insurgency led by the Lord's Resistance Army, devastated Northern Uganda's health infrastructure, displacing over 1.8 million people and disrupting access to basic services [4]. While peace has been restored since the mid-2000s, the region remains burdened by chronic poverty, underdeveloped infrastructure, and institutional mistrust.

Pader District, located in Uganda's Acholi sub-region, has a population of over 200,000 people living in dispersed rural villages with limited access to health resources [5]. The district continues to experience some of the poorest health outcomes in Uganda. Malaria remains hyperendemic and is the leading cause of morbidity and mortality among pregnant women and children under five [5]. Other infectious diseases, including pneumonia, diarrheal illnesses, tuberculosis, and HIV, persist alongside a rising burden of non-communicable diseases (NCDs) such as hypertension and diabetes [5]. Health facilities are sparsely distributed, under-resourced, and often located far from communities, limiting timely access to care [6,7].

Community health workers (CHWs) are lay health workers who deliver health promotion, preventive services, and linkages to formal care. They are typically members of the communities they serve. In Uganda, CHWs are primarily organized as Village Health Teams (VHTs) under the Ministry of Health community health strategy. This study focuses on CHWs involved in the Laro Kwo Project, a program based in Pader District that has trained and equipped approximately 150 VHT members, including some individuals initially trained through the national program who subsequently joined the project.

CHWs play an essential role in extending the reach of the health system in rural and post-conflict settings. As trusted members of their communities, they serve as critical liaisons between households and formal health services, helping to bridge gaps in access and rebuild trust in a health system still recovering from the effects of conflict and chronic under-resourcing [8] Despite their central role, CHWs operate in environments marked by limited supplies, minimal supervision, and inconsistent recognition of their work [8]. While national surveys like the Uganda Demographic and Health Survey provide important district-level health data, they are limited in capturing the lived realities at the village level where CHWs deliver care and where barriers to health access are most evident [5].

This study addresses this gap by examining CHWs' perceptions of community members' health priorities, understandings, and barriers to care in Pader District, Northern Uganda. Specifically, the study aims to (1) characterize CHWs' views of local health priorities as experienced by community members, (2) describe perceived barriers to care and health-seeking behaviors among community members, and (3) generate practical, CHW-informed recommendations to strengthen community-responsive service delivery. Findings will inform the design of a community health assessment tool co-created with CHWs to support ongoing monitoring and community-responsive programming.

## Materials and methods

### Study design and setting

This qualitative descriptive study was guided by the interpretive paradigm and conducted in collaboration with the Laro Kwo Project, a community-based health initiative operating in Pader District, Northern Uganda. The Laro Kwo Project was established in 2016 through a partnership between local leaders, Ugandan health professionals, and international collaborators, with the aim of strengthening grassroots healthcare delivery in post-conflict Northern Uganda [9]. CHWs participating in the Laro Kwo Project receive standardized training aligned with Uganda Ministry of Health community health guidelines, covering basic health promotion, disease recognition, referral pathways, and recordkeeping. Participants are provided with basic diagnostic and first aid tools such as thermometers, blood pressure cuffs, and bandages, as well as supportive equipment like waterproof backpacks, gloves, and uniforms. Training is delivered in modules over several in-person group sessions, with periodic refresher trainings led by experienced CHWs under a train-the-trainer model.

Focus group discussions were conducted in six sub-counties where the Laro Kwo Project is active: Awere, Pader Town Council, Pukor, Puranga, Kilak, and Pajule. At the time of data collection, the program had trained over 150 CHWs, who were distributed across these six sub-counties. This study forms part of a larger mixed-methods evaluation of the Laro Kwo Project.

The interpretive paradigm supported examination of how CHWs understand and interpret community members' health knowledge and access to care, drawing on their routine interactions with households. The eight domains structured the focus group guide to ensure systematic coverage of both health knowledge (e.g., symptom recognition, prevention practices, and treatment beliefs) and access-related factors (e.g., availability of services, transportation, affordability, and referral pathways). Domains were selected collaboratively with Ugandan partners to align with program priorities and local health realities. Rather than serving as analytic categories, the domains functioned as organizing prompts for discussion, allowing participants to describe how community knowledge is expressed in practice and where structural constraints limit care-seeking and service use.

### Study population

CHWs were selected through purposive sampling to ensure diverse representation across gender, age, program experience, and geographic location. Recruitment was facilitated by the program coordinator and CHW leaders from each sub-county. Each sub-county represents a distinct geographic and social catchment area, with CHWs possessing varying years of experience in the program. CHWs were eligible if they were over 18 years of age and had been

active participants in the Laro Kwo Project for at least 6 months. A total of six FGDs were conducted purposively across the six program sub-counties. Data collection continued until no new ideas or themes emerged, indicating thematic saturation.

## Data collection

Six focus group discussions (FGDs) with a total of 46 participants were conducted between July 15 and 22, 2024. One FGD was held in each of the six sub-counties where the Laro Kwo Project operates. Participants in each FGD were drawn from the same sub-county to ensure localized perspectives and to facilitate open discussion among familiar peers. Each group included 7–9 participants and sessions lasted between 90 and 120 minutes.

The FGD guide was developed collaboratively by the research team in partnership with Ugandan collaborators, including CHWs and local leaders, to ensure cultural relevance and alignment with community priorities. The final version was adapted from a community health assessment tool - and continuously adapted as discussions were conducted – to focus specifically on CHWs' professional experiences and perspectives. Questions originally intended for community members were rephrased to elicit CHWs' insights on community health concerns, care-seeking behaviors, and program challenges. Open-ended probes were used to encourage reflective discussion and clarify participants' experiences. A summary of the original guide is included in S1 Appendix, and a sample of the final version, after adaptation, is included in S2 Appendix.

The FGDs were co-facilitated by the co-principal investigators—one from the United States and one from Northern Uganda—along with the Laro Kwo Project's local research coordinator, who assisted with planning discussions, recruiting CHWs, and translating during discussions. Discussions were moderated in English, with real-time translation into Acholi when needed to ensure comprehension and participant comfort. The dual composition of the facilitation team shaped both the tone and content of discussions. The local facilitators, healthcare providers who live and work in the community, helped create a sense of cultural safety and openness, while the external researcher's presence sometimes prompted more formal or aspirational responses. These commentary dynamics were discussed openly during analysis to account for potential influence on data interpretation.

Facilitators took detailed field notes and debriefed immediately after each session to verify accuracy and completeness. Although audio recording was not used due to participant preference, two note-takers recorded key quotations and contextual details during each session, and notes were cross-checked for consistency after each discussion. Moderators encouraged participation from all attendees by directly inviting quieter members to share their views and ensuring balanced contributions across genders and experience levels.

## Data analysis

Data were analyzed using a qualitative descriptive approach informed by interpretive principles [10]. Two members of the research team, the co-principal investigators (BA and APK), independently reviewed and open-coded the field notes to develop the initial codebook. Codes were assigned to meaningful phrases and refined through iterative comparison and discussion. Similar codes were grouped into categories, from which cross-cutting themes were developed inductively. Themes were organized both around the domains of the FGD guide and emergent patterns identified during analysis. Throughout the coding process, the research team maintained analytic memos to document emerging insights, interpretive questions, and reflections on potential biases. These memos captured how the team's perspectives evolved as patterns became clearer across sub-counties and participant groups. Regular discussions between the analysts (BA and APK) and the broader research team were used to compare interpretations, refine themes, and ensure that findings remained grounded in participants' voices. This reflexive process enhanced analytic depth and transparency by linking data interpretation directly to the research context and team positionality. The study methods adhered to the Consolidated Criteria for Reporting Qualitative Studies (COREQ) [11].

## Ethical considerations

This study was approved by the Yale University Institutional Review Board (IRB#2000038006) and the Gulu University Research Ethics Committee (GUREC-2024–854), and also received formal authorization from the Pader District Government Health Department.

Written informed consent was obtained from all participants before each session. Although no sensitive personal information was collected, facilitators remained attentive to potential emotional distress when discussing poverty, illness, or conflict-related hardship. Participants were reminded of their right to withdraw at any time and were informed of local health resources available through the Pader clinics.

## Results

Six focus group discussions were conducted with a total of 46 CHWs from across the six sub-counties in Pader District where the program operates. The FGDs examined their perceptions of community health concerns, barriers to care, and health-seeking behaviors. Table 1 summarizes participant characteristics. The participants ranged in age from 29 to 58 years old, and spanned the entire lifetime of the Laro Kwo Project; one group included the first CHWs to join the program in 2016, and another group included the most recent CHWs to join just six months prior. Almost one third of all CHWs within the Laro Kwo Project were represented in the focus groups, with 46 participating out of 150 total CHWs.

Participants provided detailed accounts that reflected both clinical realities and the social, cultural, and emotional dimensions of community health. From this analysis, three overarching interpretive themes emerged (Table 2): (1) Experiencing health burden and community need, (2) Barriers to access and systemic constraints, and (3) Bridging knowledge, practice, and everyday realities.

These themes capture not only what CHWs observe in their work but also how they make sense of their roles within a health system shaped by scarcity, trust, and resilience. Across all groups, CHWs spoke with both pride and frustration, describing themselves as essential yet under-supported actors at the intersection of rural community life and public health.

### Experiencing health burden and community need

Across all discussions, malaria was consistently identified as the most pervasive and urgent health issue in Pader District. CHWs noted that nearly every household had multiple malaria cases each year, describing it as an accepted but exhausting reality of daily life.

"*Everyone here has had malaria many times, even in the same year. It is just normal now*" (P1, Male, Kilak).

Beyond malaria, CHWs identified diarrheal diseases, pneumonia, tuberculosis, HIV, and intestinal worms as common challenges. Maternal and child health concerns, including frequent pregnancies, malnutrition, and complications during childbirth, were particularly acute in remote areas. Noncommunicable diseases such as hypertension, diabetes, epilepsy,

**Table 1. Focus group participant characteristics.**

| Focus group setting | Number of Participants | Number of Males (%) | Average Age (years) | Years of experience in Laro Kwo Project |
|---|---|---|---|---|
| Puranga | 7 | 6 (86) | 45.6 | 5 |
| Awere | 7 | 7 (100) | 41.7 | 6 |
| Kilak | 8 | 6 (75) | 51.4 | 8 |
| Pajule | 9 | 7 (78) | 37.8 | 0.5 |
| Pader Town Council | 7 | 3 (43) | 43.4 | 3 |
| Pukor | 8 | 6 (75) | 38.3 | 3 |

**Table 2. Main themes from FGD analysis.**

| Theme | Sub-category | Summary |
|---|---|---|
| Experiencing Health Burden and Community Need | Infectious Diseases | Malaria is hyperendemic; repeated yearly infections are common across all age groups; diarrheal disease, pneumonia, TB, HIV/AIDS also concerns |
| | Chronic Disease Screening and Management | CHWs report increasing cases of hypertension, diabetes, ulcers, and sickle cell; limited screening and management capacity. |
| | Maternal and Child Health | Frequent pregnancies, delivery complications, and child and mother under-nutrition especially in villages located far from cities. |
| | Mental Health and Conflict Legacy | Emotional distress and depression linked to post-conflict trauma; stigma limits open discussion. Limited recognition of PTSD or anxiety |
| Barriers to Access and Systemic Constraints | Geographic Barriers to Access | Villages located 5–20km from facilities; transport unaffordable; roads impassable during rains. |
| | Drug and Supply Shortages | Government health centers face regular stockouts of malaria RDTs, antimalarials, and essential drugs. |
| | Low Quality of Care | Clinics are understaffed and overburdened; patients often experience long waits and limited follow-up. Difficult to know when clinics may have supplies or not. |
| | Reliance on CHW in underserved areas | CHWs are often the only accessible providers but lack supplies, transport, and advanced clinical training. |
| Bridging Knowledge, Practice, and Everyday Realities | Health Literacy and Preventative Care | Knowledge of symptoms and prevention exists but preventive care is rarely sought; routine screening is minimal. Bed net use is inconsistent due to limited supply or alternative uses. |
| | Nutrition and Food Insecurity | Nutritional awareness exists but cannot be practiced due to poverty and limited food variety. |
| | Water Purification and Sanitation Practices | Community members know how to purify water but often do not practice it, citing inconvenience and cost. |
| | Traditional Medicine Use | Herbal remedies are used out of necessity due to lack of access to modern care; CHWs see a shift toward modern medicine when accessible. |

and cervical cancer were also mentioned with increasing frequency, though CHWs emphasized the lack of screening and awareness.

Several CHWs reflected on the emotional weight of witnessing illness they felt powerless to treat. One participant described the frustration and sadness of being trusted as a provider in the community but unequipped to adequately help. Others mentioned that the conflict period in Northern Uganda has left lingering effects on community wellbeing.

"*We see people suffering, but there is nothing we can give them*" (P5, Female, Puranga).

## Barriers to access and systemic constraints

A central theme across all FGDs was the severe difficulty of accessing timely and adequate healthcare. CHWs described long distances to primary health facilities, typically 5–7 kilometers, sometimes up to 20, and the absence of affordable transport options. Hospitals providing emergency and advanced care are located at even greater distances and are frequently inaccessible during urgent situations. Participants explained that families often lacked the financial means for transport, fuel, or mobile airtime to seek help. Poor roads, especially during the rainy season, made the journey nearly

impossible in some villages. They emphasized that many individuals are simply too ill, lack the financial means, or both, to make such journeys. Transportation options are limited, motorbikes and bicycles are scarce, and most families cannot afford the cost of fuel, transport fares, or even mobile phone airtime to request assistance. While an ambulance system was recently introduced and viewed positively by CHWs, it still requires patients to cover fuel costs, which presents a barrier for many. Poor road infrastructure further compounds the problem, with unpaved, unlit roads that become impassable during the rainy season.

"*People want care*, *but they can't afford to get there, or they are too weak to walk*" (P1, Female, Pukor).

"*If there is no one around to drive you or let you borrow a [motorbike], then there is nothing you can do*" (P3, Male, Kilak).

Access to treatment is similarly constrained. Government health centers frequently run out of artemether-lumefantrine - the recommended first-line treatment for malaria - and private pharmacies, though better stocked, are unaffordable for many. CHWs expressed concern about declining drug efficacy and poor adherence, with some patients taking incomplete courses or saving doses for future illness. Resistance to first-line treatments is also increasingly becoming evident among community members.

"*People are taking the drugs, but they don't always get better. We think the medicine is not working like it used to*" (P6, Male, Awere).

Reaching a facility, however, does not guarantee quality care. Government-run health centers offer services at no cost, but they are often under-resourced and understaffed. Essential medications are frequently out of stock, and staff shortages can lead to long wait times and overburdened healthcare workers, potentially compromising the quality of patient interactions. CHWs noted that drug shipments typically arrive only once per quarter and are rapidly depleted. In contrast, private clinics and pharmacies tend to have more reliable supplies, but their services are financially out of reach for most community members. As a result, patients may visit multiple facilities in search of treatment, or ration medications to make them last, a strategy that, while understandable, contributes to poor health outcomes and may exacerbate problems such as antimicrobial resistance.

"*The [government-run] clinics only restock every 3 months, but they run out of medicines after two weeks*" (P7, Male, Pader Town Council).

"*Some have walked half a day to the clinic only to be turned away*" (P4, Male, Pajule).

Many CHWs described frustration when community members lost trust in them due to these systemic barriers. They reflected that being recognized as a health provider but lacking the resources to meet expectations left them feeling responsible for failures beyond their control. This sense of limitation and moral tension emerged across groups, revealing how structural barriers also affect CHWs' confidence and relationships with their communities.

"*People like me because [of my position], but they get angry with me if I cannot help or run out of supplies*" (P7, Male, Awere).

### Bridging knowledge, practice, and everyday realities

CHWs reported that health knowledge within communities varied widely, with limited awareness of preventive care and widespread delays in seeking formal healthcare services. Many community members reportedly wait until illness becomes

severe before visiting a clinic, often bypassing early intervention opportunities. Preventive services, such as blood pressure monitoring or cancer screening, are rarely sought or available outside district hospitals.

"*People only go [to the clinic] when it is already bad. They don't check their health unless something is wrong*" (P3, Female, Pader Town Council).

Even when individuals possess some knowledge about disease prevention, economic hardship and systemic limitations often prevent them from acting on this knowledge. CHWs described a consistent disconnect between knowledge and practice. For example, while many families are aware of the importance of nutrition, balanced diets are often unattainable due to food insecurity and poverty. Diets are dominated by inexpensive staples such as maize and beans, with limited access to animal protein or fresh produce. Seasonal shortages further restrict food availability, and some households intentionally reduce their food intake in order to sell more crops to cover school fees or healthcare costs.

"*They know they should eat a balanced diet, but they just eat what is available*" (P8, Female, Pajule).

Despite high community awareness of malaria symptoms and the importance of early treatment, CHWs described frequent barriers to effective prevention and care. Insecticide-treated bed nets are often insufficient in quantity, distributed infrequently, or repurposed for other uses such as fishing. Proper use and installation are inconsistent, with minimal guidance provided. Some CHWs also reported that discomfort caused by treated nets discourages regular use.

"*Most of [us] don't use the nets; the chemicals hurt [our] heads and [we] cannot sleep*" (P5, Male, Pukor).

Similar gaps were observed in water purification and hygiene practices. While many community members understand the importance of purifying drinking water, they often view the process as unnecessary or burdensome. CHWs explained that boiled or treated water is seen as an extraneous step rather than a necessity. Most households rely on wells, boreholes, or untreated surface water, with piped water available only in towns and bottled water largely unaffordable.

"*People know how to purify water, but they say it's a waste of time*" (P6, Male, Puranga).

Traditional medicine use was widespread, often driven by necessity rather than cultural preference. CHWs described the use of herbal remedies such as blackjack flower, neem leaves, and mango bark to treat common ailments like malaria, gastrointestinal issues, and wounds. While some CHWs acknowledged their potential therapeutic value, they also expressed concern that these practices can delay access to modern healthcare or result in adverse effects when used improperly. Nonetheless, participants noted a gradual shift toward biomedical care, especially when CHWs are able to offer guidance and referral.

"*[We] only go to the healer if the clinic is too crowded or too far*" (P1, Female, Pader Town Council).

CHWs reported that community members frequently approached them as initial points of contact for basic care and health education. Participants described being asked for advice, blood pressure checks, and symptom interpretation, despite often lacking the training or resources to provide direct treatment. Several reflected that this role brings both pride and pressure, balancing community expectations with limited authority and support from the formal health system.

"*I am respected in my village, and I like that people come to me [for help]*" (P6, Male, Puranga).

## Discussion

Through CHWs' accounts, the results illuminate how community members understand local health priorities and recognize barriers to care, including constraints related to distance, cost, and availability of services. The findings highlight how CHWs navigate between their communities and a health system characterized by high disease burden, limited access to services, and persistent resource constraints. Across themes related to community health burden, healthcare access and quality, and gaps between health knowledge and practice, CHWs described both the challenges they encounter and the strategies they use to support households despite constrained resources.

Malaria remains a prominent indicator of health system limitations in the study setting. As CHWs reported, nearly every household experiences repeated infections each year, reflecting persistent transmission alongside gaps in prevention and treatment capacity [5]. The availability of diagnostic and treatment tools is inconsistent, and irregular bed net distribution undermines preventive efforts. These challenges align with literature describing the difficulty of sustaining malaria control in settings with constrained supply chains and surveillance systems [12, 13]. In Pader District, CHWs described how treatment shortages and perceived treatment failure often generate frustration within communities, with CHWs frequently serving as the most visible representatives of the health system at the community level. This dynamic highlights the strain placed on CHWs when systemic limitations affect their ability to meet community expectations.

The growing visibility of NCDs such as hypertension and diabetes adds a further layer of complexity. As across much of sub-Saharan Africa, Pader now faces a dual disease burden [14]. However, the transition is complicated by limited diagnostic capacity, scarce chronic disease management infrastructure, and competing demands on households already struggling to afford daily necessities [15]. For CHWs, this evolving landscape expands their responsibilities without expanding their authority or resources. Their accounts illustrate how NCDs expose the limits of vertical, disease-specific programming and highlight the need for integrated, community-based approaches that acknowledge the realities of everyday survival [15].

Although many of the barriers identified in this study, including medicine stockouts, transport limitations, and inadequate supervision, mirror challenges reported in other low-resource settings, their manifestation in Pader District reflects long-standing structural and geographic constraints. [2,16]. Limited health infrastructure, uneven distribution of facilities, and poor road networks continue to restrict access to care across dispersed rural communities [6, 17] In this context, CHWs function not only as service extenders but as critical links between communities and an under-resourced health system, facilitating access, information exchange, and referral [18]. Their roles highlight both the potential and the limitations of CHW-led models, in which strong community relationships coexist with persistent systemic gaps [16,18].

The barriers CHWs described—distance, transport costs, and inadequate facilities—persist not only because of material scarcity but also because of structural inequities in how health systems evolve after conflict. Peripheral regions such as Pader have historically received fewer investments in infrastructure and human resources [2,7, 19]. In this environment, CHWs' credibility and motivation depend on their ability to deliver visible results, yet their means to do so are constrained [16,18]. When supplies run out or referrals fail, community trust may weaken, placing CHWs in a position of heightened moral accountability [20,21]. This tension between expectation and capacity reflects a form of structural vulnerability, in which frontline workers bear the social and emotional consequences of systemic deficiencies [22].

The mismatch between health knowledge and practice among community members highlights the importance of CHWs in bridging the gap between biomedical guidance and everyday socioeconomic constraints [18,23]. Participants described how community members often understood prevention messages but were unable to act on them due to poverty, food insecurity, and competing survival needs. In this context, CHWs described adapting health advice to local realities, prioritizing feasible actions, and helping households navigate limited options within constrained circumstances. Their insights underscore that health literacy alone is insufficient to produce behavioral change without structural support and that CHWs play a key role in translating health knowledge into practical, context-appropriate guidance [24]. These accounts

illustrate how health behavior in resource-constrained settings reflects an ongoing balance between ideal practices and achievable daily choices.

CHWs in this study described a tension between being highly trusted in their communities and working with limited resources and authority. Trust itself was not described as a burden. Instead, participants noted that trust often comes with heightened expectations for assistance, which can be difficult to meet when medicines, diagnostics, transport support, and supervisory backing are inconsistent. Similar dynamics have been observed in other community health programs, where CHW motivation is shaped by intrinsic purpose, peer recognition, and perceived fairness [16, 18] In Pader, CHWs described how chronic shortages, limited recognition, and few opportunities for advancement can strain motivation over time. At the same time, many emphasized a strong sense of purpose and solidarity with peers, while noting the risk of fatigue when expectations remain high but support remains limited.

The implications of these findings relate directly to improving alignment between local health knowledge and access to care. CHWs described communities with substantial understanding of health priorities but limited ability to act on this knowledge due to gaps in service availability, transportation, and system support [24] Strengthening community health in this context therefore requires both material inputs and supportive systems that enable CHWs to respond effectively to identified priorities. Ensuring reliable supply chains for diagnostics and essential medicines remains foundational, particularly in geographically isolated areas such as Pader District [2]. Similarly, providing basic transport support, including bicycles or modest travel stipends, could improve CHWs' ability to reach households facing physical barriers to care [16]. Digital tools may support reporting and referrals, but only if accompanied by consistent supervision and technical support [25]. Non-material interventions, including regular supportive supervision, recognition of CHW contributions, and structured opportunities for input into local planning, are also important for sustaining CHW engagement and effectiveness [26].

Together, these findings emphasize that improving community health outcomes depends not only on increasing health knowledge but on strengthening the systems that allow that knowledge to be translated into timely and accessible care, with CHWs playing a central coordinating role.

This study has several strengths. It captures diverse CHW experiences across six sub-counties, providing a rich, grounded understanding of the interface between community health and systemic constraint. The participatory design, involving Ugandan co-investigators and local facilitators, enhanced cultural relevance and ethical integrity. The inclusion of CHW voices throughout analysis ensures that recommendations are rooted in local experience rather than external assumption.

However, some limitations must be acknowledged. Findings are specific to Pader District and may not generalize to other contexts in Uganda. The use of focus groups may have introduced social desirability bias, and CHWs' views may not represent those of all community members, particularly those with limited contact with health services. Nonetheless, these insights provide an essential window into the lived realities of frontline health workers in post-conflict Northern Uganda and offer practical directions for strengthening community-based care in similar settings.

## Conclusion

This study highlights how CHWs in Pader District describe local health priorities and barriers to care in the communities they serve. Participants reported a high burden of infectious and non-communicable diseases alongside persistent constraints on access, including long travel distances to facilities, medicine stockouts, and limited transportation and infrastructure. CHWs also described substantial community health knowledge, but emphasized that financial hardship and service limitations often prevent that knowledge from translating into preventive practices and timely care-seeking.

These findings suggest that improving community health outcomes requires better alignment between local health knowledge and reliable access to services. Practical program implications include strengthening supply chains for essential diagnostics and medicines, supporting transport for referrals and household outreach, and providing regular supportive supervision. CHWs also emphasized the importance of consistent stipends, recognition, and structured opportunities to

provide input into program planning to ensure interventions remain grounded in local realities. Future directions should focus on implementing and evaluating scalable strategies identified by CHWs, such as mobile reporting tools, improved referral tracking, and peer mentorship, and examining their effects on service access, CHW engagement, and community health outcomes.

## Supporting information

**S1 Appendix. Original focus group guide.**
(DOCX)

**S2 Appendix. Adapted, final focus group guide.**
(DOCX)

**S1 Checklist. Inclusivity in global research.**
(DOCX)

## Acknowledgments

I would like to sincerely thank Dr. Daniel Ebbs, who graciously invited me to join his research team and connected me his incredible collaborators in Uganda, for his invaluable guidance and feedback throughout this project. I would also like to extend much gratitude to DHO Dr. Oyoo Benson, Bosco, David, Jacob, as well as the rest of the NUMEM team and my friends in Pader, none of this work would have been possible without their important input and hospitality every day.

## Author contributions

**Conceptualization:** Brett R. Albee, Patrick Kasagara Atiya, Felix Bongomin, Daniel Ebbs.

**Data curation:** Brett R. Albee, Patrick Kasagara Atiya, Otema Denish, Olanya Denish, Isaac V. Faustino, Shayna D Cunningham, Daniel Ebbs.

**Formal analysis:** Brett R. Albee, Patrick Kasagara Atiya, Isaac V. Faustino, Dhatri Abeyaratne, Shayna D Cunningham, Rogie Royce Carandang, Felix Bongomin, Daniel Ebbs.

**Funding acquisition:** Daniel Ebbs.

**Investigation:** Brett R. Albee, Patrick Kasagara Atiya, Otema Denish, Olanya Denish, Rogie Royce Carandang, Felix Bongomin, Daniel Ebbs.

**Methodology:** Brett R. Albee, Patrick Kasagara Atiya, Olanya Denish, Shayna D Cunningham, Rogie Royce Carandang, Felix Bongomin, Daniel Ebbs.

**Project administration:** Brett R. Albee, Patrick Kasagara Atiya, Otema Denish, Olanya Denish, Felix Bongomin, Daniel Ebbs.

**Resources:** Brett R. Albee, Patrick Kasagara Atiya, Felix Bongomin, Daniel Ebbs.

**Supervision:** Olanya Denish, Shayna D Cunningham, Felix Bongomin, Daniel Ebbs.

**Writing – original draft:** Brett R. Albee, Patrick Kasagara Atiya, Daniel Ebbs.

**Writing – review & editing:** Brett R. Albee, Patrick Kasagara Atiya, Dhatri Abeyaratne, Shayna D Cunningham, Rogie Royce Carandang, Felix Bongomin, Daniel Ebbs.

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
