## [Decision Letter · Decision Letter 0]

17 Oct 2025

PGPH-D-25-02643

Exploring Local Health Knowledge and Access: Focus Group Findings from Community Health Workers in Pader District, Uganda

Dear Dr. Ebbs,

Thank you for submitting your manuscript to PLOS Global Public Health. After careful consideration, we feel that it has merit but does not fully meet PLOS Global Public Health’s publication criteria as it currently stands. Therefore, we invite you to submit a revised version of the manuscript that addresses the points raised during the review process.

We look forward to receiving your revised manuscript.

Kind regards,

Andrew Kazibwe, MBChB

Academic Editor

Journal Requirements:

2. Please send a completed 'Competing Interests' statement, including any COIs declared by your co-authors. If you have no competing interests to declare, please state "The authors have declared that no competing interests exist". Otherwise please declare all competing interests beginning with the statement "I have read the journal's policy and the authors of this manuscript have the following competing interests:"

3. Please amend your detailed Financial Disclosure statement. This is published with the article. It must therefore be completed in full sentences and contain the exact wording you wish to be published.

1. Please clarify all sources of financial support for your study. List the grants, grant numbers, and organizations that funded your study, including funding received from your institution. Please note that suppliers of material support, including research materials, should be recognized in the Acknowledgements section rather than in the Financial Disclosure.

2. State the initials, alongside each funding source, of each author to receive each grant. For example: "This work was supported by the National Institutes of Health (####### to AM; ###### to CJ) and the National Science Foundation (###### to AM)."

3. State what role the funders took in the study. If the funders had no role in your study, please state: “The funders had no role in study design, data collection and analysis, decision to publish, or preparation of the manuscript.”

4. We have noticed that you have uploaded Supporting Information files, but you have not included a list of legends. Please add a full list of legends for your Supporting Information files after the references list.

Reviewers' comments:

Reviewer's Responses to Questions

**Comments to the Author**

1. Does this manuscript meet PLOS Global Public Health’s publication criteria?

Reviewer #1: Yes

Reviewer #2: Yes

2. Has the statistical analysis been performed appropriately and rigorously?

Reviewer #1: Yes

Reviewer #2: N/A

3. Have the authors made all data underlying the findings in their manuscript fully available (please refer to the Data Availability Statement at the start of the manuscript PDF file)?

Reviewer #1: Yes

Reviewer #2: Yes

4. Is the manuscript presented in an intelligible fashion and written in standard English?

Reviewer #1: Yes

Reviewer #2: Yes

Reviewer #1: The perspectives of community health workers regarding care barriers, community health needs, and priorities for enhancing CHW-led service delivery are examined in this study. My observations are given below:

(1) The study title is okay.

(2) The abstract needs minor revision. The abstract is thorough but excessively lengthy, which detracts from its clarity and conciseness. Although the main goals, procedures, and conclusions of the research are explained, readers may become overwhelmed by the extensive information on domains and obstacles. Think about focusing more intently.

(3) The introduction also needs minor revision. The study is clearly situated within rural and post-conflict health disparities thanks to the introduction, which is well-structured and gives a thorough overview of the contextual challenges in Northern Uganda. The case for concentrating on community health workers (CHWs) is strong, highlighting both the knowledge gap about their viewpoints and their crucial role. However, the introduction might be improved by emphasizing any theoretical frameworks directing the investigation and more clearly connecting the cited literature to the goals of the study. Contextual clarity would also be improved with a more explicit explanation for the choice of Pader District in particular.

(4) The methods section also needs improvements. The Methods section is thorough and exhibits ethical compliance, cultural sensitivity, and meticulous planning. Contextual richness is strengthened through the use of diverse focus groups and purposive sampling. There are a few drawbacks, though: the dual-language translation procedure may compromise data consistency, and the dependence on program-facilitated recruitment may introduce selection bias. Furthermore, although IPA is appropriate, inter-coder reliability metrics are not clearly described in the coding procedure description. Methodological rigor would be further improved by greater openness about reflexivity and possible researcher influence on participants.

(5) The results section also needs revision. A thorough, in-depth description of the experiences and community health issues faced by CHWs is given in the results section, which also highlights a number of structural and social factors. Nuanced themes are produced by the appropriate use of IPA and FGD data. Nevertheless, it is challenging to determine the relative importance of various factors due to the presentation's excessive descriptiveness and lack of synthesis or prioritization of findings. quantitative citations (e.g. G. "8 6 (75) 38.3 3") are ambiguous and need to be clarified. Including visual summaries or cross-theme comparisons could improve the results' readability and clarity.

(6) The discussion section also needs revision. The discussion is thorough and organized, successfully tying together the body of existing literature and CHWs' experiences with systemic health issues. The burden of infectious and non-communicable diseases, structural obstacles, and CHW resilience are all prominently highlighted. Nevertheless, the section is primarily descriptive and offers little critical examination of contextual subtleties or causal mechanisms. Some assertions, like the decline in trust brought on by a lack of resources, would benefit more from triangulated evidence. Furthermore, the conversation could more clearly distinguish Pader District-specific findings from more general sub-Saharan patterns and examine possible policy ramifications in a more analytical rather than prescriptive way.

(7) The references are ok. Please correct it by following the journal’s guidelines. The tables and figures should be prepared following standard guidelines.

Reviewer #2: This manuscript addresses a critical topic in community health and provides valuable insights into the lived realities of community health workers (CHWs) in post-conflict Northern Uganda. The study is well-contextualized, ethically grounded, and relevant for health systems strengthening. However, while it claims to employ Interpretive Phenomenological Analysis (IPA), the methodology and results align more closely with a qualitative descriptive design. The focus group format, structured guide, and domain-driven analysis limit interpretive depth. Greater attention to reflexivity, analytic transparency, and phenomenological interpretation is required to align with the stated methodological orientation.

The paper should be reframed as a qualitative descriptive study informed by interpretive principles. The authors should expand interpretive depth, include reflexivity, provide analytic traceability (e.g., codebook summary), and adjust tone for inclusivity and neutrality.

**Do you want your identity to be public for this peer review?** For information about this choice, including consent withdrawal, please see our Privacy Policy

Reviewer #1: **Yes:**  Gyanesh Kumar Tiwari

Reviewer #2: No

---

## [Decision Letter · Decision Letter 1]

6 Jan 2026

PGPH-D-25-02643R1

Exploring Local Health Knowledge and Access: Focus Group Findings from Community Health Workers in Pader District, Uganda

Dear Dr. Ebbs,

Thank you for submitting your manuscript to PLOS Global Public Health. After careful consideration, we feel that it has merit but does not fully meet PLOS Global Public Health’s publication criteria as it currently stands. Therefore, we invite you to submit a revised version of the manuscript that addresses the points raised during the review process.

Whereas your manuscript meets most of the journal's publication criteria, there are comments that ought to be addressed to make the manuscript suitable for publication. We noted some inconsistencies between the manuscript title and contents, from a qualitative perspective, which require your attention. Please refer to reviewer comments and attachment for details.

We look forward to receiving your revised manuscript.

Kind regards,

Andrew Kazibwe, MBChB, MMED

Academic Editor

Journal Requirements:

Additional Editor Comments (if provided):

Comments

Title:

The title of the abstract is well-written but lacks clarity on some key concepts.

1. Please clarify which knowledge was explored in this study? For example, knowledge of the common health problems, causes, prevention, access to care, etc.?

2. How was knowledge explored? Were you finding out how CHWs understand the common health problems, their causes, prevention, and management? Or did the study explore what CHWs know about the community’s knowledge and access to health services? In other words, was the study exploring knowledge among CHWs or knowledge among communities where CHWs come from?

Abstract

1. Lines 3-4: Clarify if community health workers in this study include village health teams (VHTs), community health extension workers (CHEWs), or both

2. Line 9: If the title is focusing on local health knowledge and access, should the methodology be focused on exploring perspectives and experiences? How about if the study had focused on exploring the CHW’s understanding of the local health problems and access to healthcare services for the identified health problems?

3. Line 10: Indicate the period during which the group discussions were conducted.

4. Line 11: How did you arrive at the eight domains? Are they adapted from another source? If so, then specify the source. If the researchers developed the domains, please indicate the criteria used to create them. Given that the FGD was developed based on the domains, should we take it that the inquiry was based on a priori themes?

5. Line 15: How are the three themes related to the eight domains?

6. Line 17-18: Maternal and child health seems so broad. Would have been good to separate and show the clear health problem e.g., diarrheal disease among children or complications of childbirth, etc.

7. Line 18: Was the community concerned about the high number of people with NCDs or the rising burden of NCDs?

8. Line 21: What does strong awareness mean?

9. Line 22-28: Some findings may be left out since they are not related to the title of this manuscript.

10. Line 25-28: The statement “…CHWs’ perspectives provide valuable insight into the strengths and shortcomings of community health delivery in post-conflict Northern Uganda…” is not related to the title of the manuscript. Whereas the title is focused on the local health knowledge and access, the conclusion is on the strengths and shortcomings of community health delivery.

11. Line 27-28: These recommendations should be harmonized with the key findings and title of the manuscript. For example, the key findings included malaria, MCH, and NCDs as key health problems. What is the recommendation on these? Secondly, the title of this manuscript focuses on local health knowledge. What is the conclusion and recommendation on this?

Introduction

1. Line 37-40: Provide citation (source of information) for the statements.

2. Line 55: Define what community health workers are. Show reference to WHO and the National policies. In the case of Uganda, specify if you are including VHTs and CHEWs among CHWs.

3. Line 67: If the gap identified in line 63 is limited information on how CHWs perceive the community health priorities, then how will filling this gap result into identification of local health priorities? Harmonize what you mean by perception as a gap and identification as an objective.

Materials and Methods

1. Line 79: How long was the training for the CHWs? Are the training tools validated and approved by the Ministry of Health? What is the basic content of the training? How does this training affect to the perception of CHWs on the common health priorities? How many CHWs have been trained? Are there refresher trainings for those who were trained over 2 years ago?

2. Line 67: How are the CHWs trained and supported by the Laro Kwo project different from those not trained by the project? Are the trainings and support by the project likely to result into difference in the understanding of the local health problems?

3. What support does the project provide to the CHWs? How does the support relate to the barriers to community health care?

4. Line 83: How was the selection of the CHWs? What are the basic qualifications of persons selected and trained as CHWs? How many CHWs are trained per village? Are there CHWs who were not trained by the project?

5. Line 85: “…The in-depth understanding of CHWs' roles and experiences within their communities and the broader health system context…” does not match well with the focus of the study. It would be better to describe the methodology in relation to the title. For example, “….in-depth understanding of CHWs' knowledge of the community health problems…”

6. Line 88: It remains unclear how the eight domains were used to explore the local health knowledge and access.

7. Lines 110-112: “…The final version was adapted from an earlier community health assessment tool - and continuously adapted as discussions were conducted – to focus specifically on CHWs’ professional experiences and perspectives…”

• The final adapted focus group guide in Appendix 2 (https://www2.cloud.editorialmanager.com/pgph/download.aspx?id=631060&guid=099085a8-5583-47c3-88d7-28fc6b6441cb&scheme=1) is more of an individual interview guide, not a guide for FGD. Many of the questions are closed-ended and directed at an individual. For example, have you ever had a pap smear? Mammogram? Colonoscopy? Have you had an eye exam? Please clarify whether this is the tool that was used in the FGDs and how the qualitative data was collected using this tool.

• Using this adapted assessment tool, was this a quantitative or qualitative study?

8. Line 118-120: “…The FGDs were co-facilitated by the co-principal investigators—one from the United States and one from Northern Uganda—alongside the Laro Kwo Project’s local research and program coordinator, who assisted with planning, logistics, and translation…” What does “alongside” mean? Did the program coordinator participate in the FGDs? If so, how could his/her participation influenced the responses from the CHWs?

Results

1. Line 161-163: “…The FGDs examined their perceptions of CHWs on community health concerns, barriers to care, and health-seeking behaviors…” This is not consistent with the research gap described in the introduction section of the manuscript. The gap was stated as “perception of CHW on community health priorities”, the aim of the study was stated as “to identify community health priorities”, and now the result section shows that “the FGDs examined CHWs’ perceptions on community health concerns”. I don’t think that these praises have the same meaning; therefore, they should be harmonized.

2. Why were male CHWs participating in FGDs more than females in 5 out of 6 sub-counties? What proportion of the 150 CHWs trained by the Laro Kwo project are females? Were male CHWs purposively targeted during the selection of the participants for the FGDs? You need to add this explanation to the narrative.

3. Line 172-174: These three themes stated in the results section are not the same as those in the abstract, i.e., (1) community health burden and disease priorities, (2) healthcare quality and access, and (3) community knowledge and health practices. Please harmonize.

4. Line 179: In Table 2:

• The word “hyperendemic” cannot be reported by a CHW in a qualitative study. Maybe you meant “malaria is very common throughout the year.”

• Why is malaria separated from other infectious diseases in the summary?

• What does “repeated yearly infections” mean? Is this a true summary or paraphrase of what the CHWs said? Perhaps the use of simpler descriptions would help, e.g., “diseases which people suffer more than once in a year.”

• Limited screening and management capacity is stated under the sub-theme of rising chronic diseases. Limited screening and management capacity is not a disease, therefore cannot be under this theme.

• Frequent pregnancies are stated under the maternal and child health sub-theme. How did CHWs raise this as a concern?

• You state that “…child and mother under-nutrition especially in rural areas…” What proportion of the Pader district population is urban and rural? Were CHWs able to distinguish that under-nutrition was more in rural areas of Pader district? You need to review the FGD notes to ensure that some of these issues were accurately captured.

• Under the mental health and conflict legacy sub-theme, you state that “…Emotional distress and depression linked to post-conflict trauma; stigma limits open discussion…” How did the CHWs mention these as community health priorities/concerns? Do you have quotations to support these issues? They sound more like issues that can only be raised by highly trained/qualified respondents like health workers. The CHWs may have mentioned issues that imply psychological distress, but concluding that emotional distress and depression, and linking them to post-conflict trauma, could be difficult in a study like this one.

• The second theme is stated as “Navigating barriers and systemic constraints.” Why was the word “navigating” used in this theme? Does it show what the CHWs are doing to address the barriers to access? Did the FGDs discuss how the community is addressing barriers? If so, can you identify the questions in the FGD guide that focused on this? This theme is also not well stated in relation to the title of the manuscript. If the focus of the manuscript is on barriers to health care access, then the theme should not be navigating barriers, but rather clearer and focused themes like cultural beliefs and practices, personal and family inadequacies, financial constraints, health system inadequacies, etc.

• The sub-theme stated as “inconsistent quality of care” is not clear. Depending on the codes generated and how the inductive data analysis process was done, clarify whether this would be better summarized as inconsistent or low quality of care? Clarify what you mean by inconsistent.

• It is stated that “Herbal remedies are used out of necessity due to lack of access to formal care.” What did you mean by “formal care”? If formal care meant modern medical care provided in health facilities, is it lacking or low? Check with the original data, codes and meaning units to ensure that this meaning was maintained.

5. Line 189-191: Provide quotations

6. Line 198-199: “Others linked these struggles to the lasting trauma of the conflict period, suggesting that grief, displacement, and hardship have left lingering effects on community wellbeing.” It is not clear how the difficulty in accessing health services is being linked to the lasting trauma of the conflict period. Was this finding well validated with strategies like audit trail, member checks and triangulation? Secondly, avoid sentiments like “these struggles” since this is the researcher’s language. Not a quotation of the participant.

7. Line 202-204: “CHWs described long distances to facilities, typically 5–7 kilometers, sometimes up to 20, and the absence of affordable transport options. Hospitals were even farther, often unreachable during emergencies.” Distinguish between the hospitals and facilities in the sentences.

8. Lines 206-216: Better to separate the narrative for these findings such that issues of transport, financial challenges, lack of medicines in health facilities etc. are narrated differently with quotations as a way of showing that these were participant voices.

9. Results Section: Provide quotations for all the results. Use a uniform and standard way of writing quotations, e.g., (Participant 3, Awere FGD) or (P6, Male, Pajule FGD), etc. Note that quotations are usually not in the narrative section. The quotation is placed below the narrative, in its own paragraph, indented beyond the main paragraph, italic, and having a citation or participant identifier to show source of the quotation. Some key results require more than one quotation. For example:

CHWs expressed concern about declining drug efficacy and poor adherence, with some patients taking incomplete courses or saving doses for future illness:

“People are taking the drugs, but they don’t always get better. We think the medicine is not working like it used to” (P3, Female, Awere FGD).

Some people in a home share medicine for one person because they do not have enough from the hospital” (P5, Male, Kilak FGD).

Structural barriers affected CHWs’ confidence and relationships with their communities.

“Sometimes people ask me if I am a doctor or a nurse, and I fail to give them a good answer” (P2, Male, Pajule FGD).

10. Line 276-277: “As trusted figures within their communities, CHWs increasingly serve as informal first points of contact for basic care and health education.” This sounds like an interpretation or discussion of results, which should not be in this section.

Discussion

1. Line 285-287: “The findings illustrate how CHWs operate within a health landscape shaped by structural fragility, poverty, and the enduring effects of conflict, yet also demonstrate their adaptability, resilience, and strong sense…” Are you sure the findings of this study demonstrate these?

2. Line 289: Why start by discussing barriers to accessing health care? Given your manuscript title, aim, and results, it would have been better to start by discussing the local health knowledge.

3. Line 291: “…their manifestation in Pader District reflects a distinct post-conflict reality.” Clarify what you mean here. Do you mean the manifestation of these challenges in Pader district are due to the post-conflict situation? These issues seem to be the general challenges facing the health sector in the rest of Uganda irrespective of the post-conflict situation.

4. Line 301-302: “The consistent availability of diagnostic and treatment tools remains elusive…” Use a simple and neutral tone. Avoid words like elusive unless they really add value to what you aim to communicate.

5. Line 304-305: “…malaria also functions as a barometer of public trust…” Use a simpler and neutral tone.

6. Line 326-328: “The mismatch between health knowledge and practice among community members further underscores the role of CHWs as navigators of both biomedical and socioeconomic realities.” Not clear. Do you mean the mismatch shows the role of CHWs? So, how does the role of CHWs solve the issues that are responsible for the mismatch?

7. Line 330: “…health literacy cannot translate into behavioral change without structural support…” Is this your opinion, or is it a known theory/phenomenon? Provide Citation.

8. Line 331-332: “The CHWs’ insights highlight that education alone cannot overcome material deprivation and that health behavior is inseparable from economic circumstance.” This study alone cannot make you arrive at this conclusion with certainty.

9. Line 335: “The dual burden carried by CHWs, being both trusted and under-resourced providers.” Is this accurate? Being under-resourced may be a burden, but should being trusted also be a burden? Please clarify.

10. Line 337: “…often absorbing community frustration…” Provide a citation for this statement. Should this study alone conclude that “CHWs are often absorbing community frustration”?

11. Line 346: The discussion seems to be focusing so much on CHWs' experience and challenges in doing their work, rather than the local health knowledge and access. My understanding is that the CHWs were only the source of the information, not the focus of this study. The discussion should focus on the key findings in relation to the title and aim of the manuscript.

12. Lines 348-350: “…attention to the social and relational dimensions of care . First, ensuring reliable supply chains for diagnostics and essential medicines remains fundamental, particularly in geographically isolated areas such as Pader District…” Which results of this study relates to social and relational dimensions of care? How do these relate with social and relational dimensions of care? How does “reliable supply chains for diagnostics and essential medicines” relate with social and relational dimensions of care?

13. Line 352-356: “…Third, digital tools have potential to support data reporting and supervision, but they must be embedded in systems that provide ongoing mentorship and technical support. Equally critical are non-material interventions: regular supportive supervision, visible recognition of CHW contributions, and formal inclusion in local health decision-making….” Which result of the study are you discussing in this paragraph?

14. Line 358-361: “…this study situates CHWs as agents of both healthcare delivery and social recovery. In Pader, they represent a locally rooted response to the long-term consequences of conflict: rebuilding trust, restoring communication between households and clinics, and embodying the promise of community resilience…” This was not the aim or objective of this study. Please explain how this study situates CHWs as agents of both healthcare delivery and social recovery. Was this a finding of the study? If not, then provide a citation.

15. Line 378-379: “Community health workers in Pader District play a vital role in addressing persistent health challenges in rural Northern Uganda…” This was not a finding nor a focus of the study. Therefore, consider removing this statement from the conclusion.

16. Line 383-392: The recommendations should be focused on local health knowledge and access. CHWs can be part of that.

Reviewers' comments:

Reviewer's Responses to Questions

**Comments to the Author**

Reviewer #1: All comments have been addressed

Reviewer #3: (No Response)

publication criteria?

Reviewer #1: Yes

Reviewer #3: Partly

3. Has the statistical analysis been performed appropriately and rigorously?

Reviewer #1: Yes

Reviewer #3: Yes

4. Have the authors made all data underlying the findings in their manuscript fully available (please refer to the Data Availability Statement at the start of the manuscript PDF file)?

Reviewer #1: Yes

Reviewer #3: Yes

5. Is the manuscript presented in an intelligible fashion and written in standard English?

Reviewer #1: Yes

Reviewer #3: Yes

Reviewer #1: I carefully and intently read the manuscript. The questions posed in the initial article have been addressed by the writers. The manuscript is now easier to read and more thorough. It is now more technically sound. The manuscript can add to the body of literature in its current form, in my opinion.

Reviewer #3: Major comments are:

1. The focus group discussion guide used in the study and attached to the manuscript in Appendix 2 appears to be a questionnaire with many closed-ended questions for individual interviews, which raised concerns of how the qualitative data was collected. Unfortunatly, the author indicated in the methods section that the FGDs were not audio recorded and transcribed. The FGDs were recorded using handwritten notes. This makes it challenging if an audit is to be done from data collection to analysis. Therefore, the author should revise the FGD guide and only show the questions that were used in the focus group discussions.

2. There is a general inconsistence from the title, research gap, aim of the study, results, discussion and conclusion. The focus of the study changes from local health knowledge and acess, to common health concerns, identifying health priorities, and CHW roles and challenges in the post conflict context. The discussion, conlusion and recommendations focus more on CHW roles and challenges.

3. In the results section of the manuscript, the quotations are not adequate i.e. many key results do not have quotations. Where quotations are provided, they are not formatted according to standard guidelines like COREQ guidelines. For example, most quotations don't have identification like participant number and FGD location, and not presented in a separate indented paragraphs.

Detailed comments are in the attached Manuscript Reviewer's comments.

**Do you want your identity to be public for this peer review?** For information about this choice, including consent withdrawal, please see our Privacy Policy

Reviewer #1: **Yes:**  Gyanesh Kumar Tiwari

Reviewer #3: No

---

## [Editor Report · Decision Letter 2]

9 Feb 2026

Community Heath Knowledge and Access to Care in Post-Conflict Northern Uganda: Perspectives of Community Health Workers in Pader District

PGPH-D-25-02643R2

Dear Dr. Ebbs,

We are pleased to inform you that your manuscript 'Community Heath Knowledge and Access to Care in Post-Conflict Northern Uganda: Perspectives of Community Health Workers in Pader District' has been provisionally accepted for publication in PLOS Global Public Health.

Best regards,

Andrew Kazibwe, MBChB, MMED

Academic Editor

Review entire manuscript and appendices to ensure correct spellings and grammar. For example, "tradicional" in Appendix 2.